# Combinatorial Effects of CPP-Modified Antimicrobial Peptides: Synergistic and Additive Interactions Against Pathogenic Bacteria

**DOI:** 10.3390/ijms26135968

**Published:** 2025-06-21

**Authors:** Oxana V. Galzitskaya, Sergey V. Kravchenko, Sergei Y. Grishin, Alena P. Zakhareva, Leila G. Mustaeva, Elena Y. Gorbunova, Alexey K. Surin, Viacheslav N. Azev

**Affiliations:** 1Gamaleya Research Centre of Epidemiology and Microbiology, Moscow 123098, Russia; 2Institute of Theoretical and Experimental Biophysics, Russian Academy of Sciences, Pushchino 142290, Russia; 3Institute of Environmental and Agricultural Biology (X-BIO), Tyumen State University, Tyumen 625003, Russia; svkraft@yandex.ru (S.V.K.); heggiesan@gmail.com (A.P.Z.); alan@vega.protres.ru (A.K.S.); 4Institute of Protein Research, Russian Academy of Sciences, Pushchino 142290, Russia; syugrishin@gmail.com; 5The Branch of the Institute of Bioorganic Chemistry, Russian Academy of Sciences, Pushchino 142290, Russia; mustaeva@rambler.ru (L.G.M.); eyugorbunova@rambler.ru (E.Y.G.); viatcheslav.azev@bibch.ru (V.N.A.); 6State Research Center for Applied Microbiology and Biotechnology, Obolensk 142279, Russia

**Keywords:** antimicrobial peptides, amyloidogenic regions, cell-penetrating peptide, TAT fragment, Antp fragment, peptide synergy, MRSA

## Abstract

The development of novel antimicrobial peptides (AMPs) with broad-spectrum activity represents a promising strategy to overcome multidrug resistance in pathogenic bacteria. In this study, we investigated the antimicrobial activity of three designed peptides—R44K^S^*, V31K^S^*, and R23F^S^*—engineered to incorporate an amyloidogenic fragment from the S1 protein of *Staphylococcus aureus* and one or two cell-penetrating peptide (CPP) fragments to enhance cellular uptake. The antimicrobial efficacy of these peptides and their combinations was assessed against *Pseudomonas aeruginosa*, *Escherichia coli*, *Staphylococcus aureus*, methicillin-resistant *S. aureus* (MRSA), and *Bacillus cereus*. The results demonstrated that all three peptides exhibited significant antibacterial activity in a concentration-dependent manner, with R44K^S^* being the most potent. Peptide combinations, particularly V31K^S^*/R23F^S^* and R44K^S^*/V31K^S^*, showed enhanced inhibitory effects and reduced minimum inhibitory concentrations (MICs), suggesting synergistic or additive interactions. Fractional inhibitory concentration index (FICI) analysis confirmed that most combinations exhibited synergy or additive effects. These findings highlight the potential of CPP-modified peptides as antimicrobial agents and underscore the importance of optimizing peptide combinations for therapeutic applications.

## 1. Introduction

Antimicrobial peptides (AMPs), with their broad-spectrum activity and ability to form pores in microbial membranes, are promising candidates for overcoming multidrug resistance [1,2,3,4,5]. These effects are mediated through several distinct mechanisms, including barrel-stave, toroidal, and carpet-like models of membrane interaction [6]. Peptides offer desirable properties for therapeutics, such as high target specificity, biocompatibility, and cost-effective production, but their clinical application is hindered by poor delivery to target sites [7,8,9,10]. Synthetic antimicrobial peptides offer a fast and effective approach to the development of potent antibacterial agents [11,12,13,14].

In recent years, the combination of AMPs with cell-penetrating peptides (CPPs) has emerged as a potential means to enhance their therapeutic efficacy. CPPs are short peptides known for their ability to facilitate the intracellular delivery of various cargoes, including peptides, nucleic acids, and drugs [15]. CPPs are capable of crossing cellular membranes either through direct membrane translocation or by endocytosis, typically mediated by their interactions with negatively charged components of the membrane, including glycosaminoglycans and phospholipids [16,17]. The antennapedia peptide (RQIKIWFQNRRMKWKK) and TAT fragment (RKKRRQRRR) were among the first CPPs identified in the 1990s and have since been widely used as membranotropic molecules, including in the design of antimicrobial peptides [18,19]. Exploring CPPs as components of antimicrobial peptides remains an area of interest [20,21]. The fusion of CPPs with AMPs holds the promise of not only increasing their stability and bioavailability but also enhancing their ability to penetrate bacterial cells [22]. The combination of CPP and AMP functionalities within hybrid peptides may enhance both uptake efficiency and antimicrobial potency by facilitating membrane binding, translocation, and internal targeting. This modular design can result in synergistic effects, whereby CPP-mediated delivery increases the local concentration of AMP fragments at or within the bacterial membrane, amplifying their disruptive action [23].

Moreover, the incorporation of amyloidogenic fragments, which are capable of inducing misfolding and aggregation, offers an additional mechanism to disrupt bacterial cellular integrity and function [24,25,26,27,28]. Specifically, we synthesized three peptides—R44K^S^*, V31K^S^*, and R23F^S^*—each containing an amyloidogenic fragment based on the amino acid sequence of the S1 protein from *Staphylococcus aureus* [29]. R23F^S^* was engineered with a single TAT fragment, V31K^S^* incorporated a single Antennapedia peptide (Antp), and R44K^S^* was designed as a dihybrid peptide integrating both CPPs (Figure 1).

One particularly effective strategy involves the synergistic combination of AMPs. Synergistic interactions between antimicrobial compounds can enhance bacterial eradication, reduce the required dosages of individual agents, and minimize the risk of resistance development [31,32]. Several studies have demonstrated that the combination of multiple AMPs or AMPs with antibiotics can enhance the effectiveness of individual antimicrobial agents against resistant bacterial strains and expand their spectrum of antimicrobial activity [33,34,35,36].

The aim of this study was to investigate the synergistic effects of individual antimicrobial peptides (AMPs) to enhance antibacterial efficacy. To achieve this, three previously designed AMPs (R44K^S^*, V31K^S^*, and R23F^S^*) were tested in pairwise combinations against Gram-negative (*Escherichia coli* K12, *Pseudomonas aeruginosa* ATCC 28753) and Gram-positive (*Staphylococcus aureus* 209P, MRSA ATCC 43300, *Bacillus cereus* IP 5832) bacterial strains. Fractional inhibitory concentration indexes (FICIs) were calculated to assess synergy, additive effects, lack of interaction, or antagonism between the peptides in combination.

## 2. Results

### 2.1. Combined Antimicrobial Effects of Peptides Against Pseudomonas aeruginosa

In this study, we examined the antimicrobial activity of three designed peptides—R44K^S^*, V31K^S^*, and R23F^S^*. These peptides were engineered to incorporate an amyloidogenic fragment derived from the S1 protein of *Staphylococcus aureus*, along with either one or two cell-penetrating peptide (CPP) fragments to enhance cellular uptake and bioactivity. To assess their potential for synergistic interactions, we tested the combinations of R44K^S^* with V31K^S^*, R44K^S^* with R23F^S^*, and V31K^S^* with R44K^S^* in a liquid medium against *Pseudomonas aeruginosa* (ATCC 28753 strain). By evaluating their combined effects, we aimed to determine whether the interaction of these peptides could enhance their antimicrobial efficacy beyond their individual activities, which could provide valuable insights into optimizing peptide-based therapeutic strategies.

Initially, we assessed the combined effects of peptides R23F^S^*, V31K^S^*, and R44K^S^* in a liquid medium against Gram-negative *P. aeruginosa*. The combinational effect was compared for peptide pairs in the V31K^S^*/R23F^S^*, R44K^S^*/V31K^S^*, and R44K^S^*/R23F^S^* combinations (Figure 2).

As shown in Figure 2, the peptides exhibited antimicrobial effects against *P. aeruginosa*, strain ATCC 28753. The minimum inhibitory concentration (MIC) of each individual tested peptide, at which no bacterial growth was observed, was 6 µM for R23F^S^* (Figure 2A), 12 µM for V31K^S^* (Figure 2B), and 1.5 µM for R44K^S^* (Figure 2C).

At the same time, the peptide combination V31K^S^*/R23F^S^* exhibits an enhanced inhibitory effect on the growth of *P. aeruginosa* by reducing the effective concentrations compared to the individual peptides V31K^S^* and R23F^S^*. This peptide pair at 1.5/3 µM and 3/1.5 µM completely suppresses the growth of *P. aeruginosa* (Figure 2D). The R44K^S^*/V31K^S^* and R44K^S^*/R23F^S^* combinations exhibit bactericidal effects at concentrations of 0.75/3 µM and above (Figure 2E,F).

### 2.2. Combined Antimicrobial Effects of Peptides Against Escherichia coli

To assess their potential for synergistic interactions, we tested the combinations of R44K^S^* with R23F^S^*, R44K^S^* with V31K^S^*, and R44K^S^* with R23F^S^* in a liquid medium against *E. coli* (K12 strain) (Figure 3).

As shown in Figure 3, the peptides R23F^S^*, V31K^S^*, and R44K^S^* individually exhibit significant antibacterial activity depending on their concentration. The peptide R23F^S^* inhibited the growth of the K12 strain of *E. coli* at concentrations ≥3 µM (Figure 3A). The peptide V31K^S^* displayed similar activity to R23F^S^* (Figure 3B). The peptide R44K^S^* demonstrated relatively stronger antibacterial effects compared to R23F^S^* and V31K^S^*, with 1.5 µM completely suppressing *E. coli* growth for 15 h of incubation (Figure 3C). The antimicrobial effects in peptide combinations were generally concentration-dependent. The combination V31K^S^*/R23F^S^* did not exhibit a bactericidal effect on the growth of *E. coli* culture (Figure 3D). In contrast, the R44K^S^*/V31K^S^* pair demonstrated a bactericidal effect at concentrations of 0.75/0.75 µM (Figure 3E). For the R44K^S^*/R23F^S^* pair, a trend toward growth inhibition was observed; however, no bactericidal effect was detected (Figure 3F).

### 2.3. Combined Antimicrobial Effects of Peptides Against Staphylococcus aureus

To evaluate their potential synergistic interactions, we examined the combinations of R44K^S^* with R23F^S^*, R44K^S^* with V31K^S^*, and V31K^S^* with R23F^S^* in a liquid medium against *S. aureus* (strain 209P) (Figure 4).

The peptide R23F^S^* exhibited dose-dependent antibacterial activity. Higher concentrations (≥6 µM) led to complete inhibition of *S. aureus* (strain 209P) growth (Figure 4A). V31K^S^* demonstrated weaker inhibition compared to R23F^S^*, as its bactericidal effect was observed only at concentrations ≥12 µM (Figure 4B). However, the lowest MIC value among all individually tested peptides against *S. aureus* was recorded for R44K^S^* at 3 µM (Figure 4C). The peptides in the V31K^S^*/R23F^S^* mixture exhibited a bactericidal effect at lower concentrations than each peptide individually. The combination V31K^S^*/R23F^S^* at concentrations of 1.5/3 µM and above significantly inhibited bacterial growth, indicating a potential synergistic effect (Figure 4D). The R44K^S^*/V31K^S^* peptide mixture completely inhibited bacterial growth for 15 h of co-incubation, similar to gentamicin, which suggests a synergistic interaction between these peptides (Figure 4E). At concentrations of 0.75/1.5 µM and above, the R44K^S^*/R23F^S^* peptide pair exhibited a bactericidal effect against the *S. aureus* culture (Figure 4F).

### 2.4. Combined Antimicrobial Effects of Peptides Against Methicillin-Resistant Staphylococcus aureus

To assess their potential synergistic interactions, we tested the combinations of R44K^S^* with R23F^S^*, R44K^S^* with V31K^S^*, and V31K^S^* with R23F^S^* in a liquid medium against MRSA (strain ATCC 43300) (Figure 5).

As demonstrated in Figure 5, among the individually tested peptides R23F^S^*, V31K^S^*, and R44K^S^*, the highest antimicrobial effect was observed for R44K^S^*, as it had the lowest MIC value of 6 µM against MRSA, ATCC 43300 strain (Figure 5C). Under the same conditions, the MIC values for R23F^S^* and V31K^S^* were 12 µM and 24 µM, respectively (Figure 5A,B). The antibacterial activity of the peptide pair combinations R44K^S^* with V31K^S^* and R44K^S^* with R23F^S^* was higher compared to the individually tested peptides due to a reduction in MIC values within the combinations. For the R44K^S^*/V31K^S^* combination, a bactericidal effect against MRSA was observed at concentrations of 3/0.75 µM and above (Figure 5D). Similarly, the MIC in the R44K^S^*/R23F^S^* combination was lower than the MIC values for the individual peptides and was determined to be 1.5/3 µM (Figure 5F). Only the V31K^S^*/R23F^S^* combination did not exhibit a bactericidal effect (Figure 5E).

### 2.5. Combined Antimicrobial Effects of Peptides Against Bacillus cereus

To evaluate their potential combinatory effects, we examined the combinations of V31K^S^* with R23F^S^*, R44K^S^* with V31K^S^*, and R44K^S^* with R23F^S^* in a liquid medium against *B. cereus* (IP 5832 strain) (Figure 6).

The peptides R23F^S^*, V31K^S^*, and R44K^S^* exhibited significant antimicrobial effects against *B. cereus*, as confirmed by MIC values of 3 µM, 1.5 µM, and 0.75 µM, respectively (Figure 6A–C). The bactericidal effect of V31K^S^*/R23F^S^* is observed at peptide concentrations of 1.5/3 µM and above (Figure 6D). The peptide combinations R44K^S^*/V31K^S^* and R44K^S^*/R23F^S^* exhibited bactericidal activity against *B. cereus* at peptide concentrations of 0.18/0.75 µM and above (Figure 6E,F).

### 2.6. The Results of the Fractional Inhibitory Concentration Index (FICI) Calculations for Pairwise Peptide Combinations Against E. coli, P. aeruginosa, S. aureus, MRSA, and B. cereus

The antibacterial activity of pairwise peptide combinations R23F^S^*, V31K^S^*, and R44K^S^* was evaluated using the standard microdilution method. The concentrations of the peptide combinations V31K^S^*/R23F^S^*, R44K^S^*/V31K^S^*, and R44K^S^*/R23F^S^* were tested using the checkerboard assay to determine the synergistic response of these combinations in terms of antimicrobial activity against *E. coli*, *P. aeruginosa*, *S. aureus*, MRSA, and *B. cereus* (Table 1).

The analysis of the antibacterial activity of the peptide combinations was performed by calculating the fractional inhibitory concentration index (FICI), which indicate whether a peptide combination is synergistic (FICI ≤ 0.5), additive (0.5 < FICI ≤ 1), non-interactive (1 < FICI ≤ 4), or antagonistic (FICI > 4) [37].

According to the data in Table 1, it can be concluded that most of the tested peptide pair combinations—V31K^S^*/R23F^S^*, R44K^S^*/V31K^S^*, and R44K^S^*/R23F^S^*—exhibited additive (0.5 < FICI ≤ 1) and synergistic (FICI ≤ 0.5) antimicrobial effects. For the V31K^S^*/R23F^S^* combination against *B. cereus*, the FICI value was 2, indicating no combinatory effect.

## 3. Discussions

In this study, we investigated the antimicrobial properties of three designed peptides—R44K^S^*, V31K^S^*, and R23F^S^*—which were engineered to incorporate an amyloidogenic fragment derived from the S1 protein of *Staphylococcus aureus*, along with one or two cell-penetrating peptide (CPP) fragments. The antimicrobial activity of these peptides was tested individually and in combination against a range of bacterial species, including *Pseudomonas aeruginosa*, *Escherichia coli*, *Staphylococcus aureus*, methicillin-resistant *Staphylococcus aureus* (MRSA), and *Bacillus cereus*. The findings of this study provide significant insights into the antimicrobial potential of the designed peptides R44K^S^*, V31K^S^*, and R23F^S^*, as well as their combinatorial effects against various bacteria. The observed antimicrobial activity of these peptides, particularly their effectiveness against Gram-negative and Gram-positive pathogens, aligns with previous research highlighting the potential of synthetic antimicrobial peptides (AMPs) as alternatives to conventional antibiotics [38]. The integration of amyloidogenic fragments from the S1 protein of *Staphylococcus aureus* with CPP fragments in the studied peptides appears to enhance their antibacterial efficacy, likely due to improved cellular uptake and membrane interaction, as suggested in similar studies on CPP-modified antimicrobial peptides [39].

The individual MIC values observed for R44K^S^*, V31K^S^*, and R23F^S^* demonstrate varying levels of antibacterial potency, with R44K^S^* exhibiting the highest efficacy across multiple bacterial species. This result is consistent with the presence of two CPP fragments in the peptide structure, which predominantly contain positively charged amino acid residues, facilitating better binding of R44K^S^* to bacterial membranes [29]. Antimicrobial peptides that exhibit stronger binding to bacterial membranes are generally considered highly effective, as interactions with bacterial membranes are fundamental to the primary mechanisms of AMP activity [40,41,42].

In tests against *P. aeruginosa*, the combination of V31K^S^*/R23F^S^* exhibited a significantly enhanced inhibitory effect, completely suppressing bacterial growth at lower concentrations compared to individual peptides. Similarly, the combinations of R44K^S^*/V31K^S^* and R44K^S^*/R23F^S^* demonstrated bactericidal activity at reduced concentrations, further indicating a beneficial interactive effect between these peptides. Additive effects observed in peptide combinations further support the hypothesis that specific interactions between peptides can enhance their antimicrobial properties [43].

Against *E. coli*, individual peptides exhibited a dose-dependent antibacterial effect, with R44K^S^* demonstrating the strongest activity. While V31K^S^*/R23F^S^* and R44K^S^*/R23F^S^* did not exhibit a bactericidal effect, the combination of R44K^S^*/V31K^S^* displayed enhanced activity, achieving bactericidal effects at lower concentrations than those required for the individual peptides. The absence of a bactericidal effect in the V31K^S^*/R23F^S^* and R44K^S^*/R23F^S^* combinations against *E. coli* was unexpected. A possible explanation is that the individual peptides R23F^S^*, V31K^S^*, and R44K^S^* exhibited low MIC values, leading to the selection of even lower concentration ranges for testing combinatorial effects. At relatively low concentrations, certain peptide combinations may exhibit neutral or even antagonistic interactions due to competition for membrane binding and the inability to reach all bacterial targets [44,45].

For *S. aureus*, the R44K^S^*/V31K^S^* combination exhibited a synergistic effect (FICI = 0.31), suppressing bacterial growth at lower concentrations than either peptide alone. The V31K^S^*/R23F^S^* and R44K^S^*/R23F^S^* combinations also exhibited bactericidal effects at relatively low concentrations. The ability of the V31K^S^*/R23F^S^*, R44K^S^*/V31K^S^*, and R44K^S^*/R23F^S^* combinations to exhibit bactericidal effects at relatively low concentrations suggests that these peptides could serve as viable candidates for developing new antimicrobial strategies against *S. aureus* infections [46,47].

For MRSA, the R44K^S^*/V31K^S^* and R44K^S^*/R23F^S^* combinations exhibited reduced MIC values compared to the individual peptides, reinforcing the potential benefits of AMP combinations in tackling resistant bacterial strains [48]. However, the V31K^S^*/R23F^S^* combination did not exhibit a bactericidal effect against MRSA. Overall, the V31K^S^*/R23F^S^* combination exhibited weaker antimicrobial effects compared to other combinations and against different bacterial species (Table 1), which may be attributed to the structural characteristics of the peptides. V31K^S^* and R23F^S^* each contain a single CPP fragment, whereas R44K^S^* incorporates two CPPs, potentially providing an advantage in bacterial membrane binding [29]. Consequently, it can be assumed that combinations containing R44K^S^* exhibit stronger antimicrobial effects compared to the V31K^S^*/R23F^S^* combination. Overall, the synergistic effects of peptides may be diminished against the ATCC 43300 methicillin-resistant *S. aureus* strain compared to the 209P *S. aureus* strain, primarily due to physiological and biochemical differences between these two types of strains. For example, it is known that MRSA modifies its membrane composition, notably by increasing lysyl-phosphatidylglycerol via the MprF enzyme and enhancing D-alanylation of teichoic acids through the DltABCD pathway, which reduces the membrane surface charge and decreases binding of cationic peptides [49,50]. Additionally, MRSA often alters membrane fluidity and the glycosylation profiles of wall teichoic acids, creating a more robust barrier against AMPs [51]. These adaptive mechanisms are likely responsible for the reduced efficacy of the peptide combination in MRSA compared to methicillin-sensitive *S. aureus*.

Finally, the V31K^S^*/R23F^S^* combination displayed bactericidal effects against *B. cereus* at relatively higher concentrations, whereas the R44K^S^*/V31K^S^* and R44K^S^*/R23F^S^* combinations exhibited potent bactericidal activity at significantly lower concentrations. FICI analysis confirmed that the R44K^S^*/V31K^S^* and R44K^S^*/R23F^S^* combinations resulted in additive and synergistic antimicrobial effects, respectively. Notably, the V31K^S^*/R23F^S^* combination against *B. cereus* exhibited no combinatory effect, with an FICI value of 2. These results emphasize the importance of both the rational design of antimicrobial peptides and the selection of AMP combinations to enhance their antimicrobial efficacy [52,53,54].

However, it is important to acknowledge certain limitations of the present study. In particular, the experimental data were obtained from only two biological replicates, which constrains the statistical power of the conclusions, especially those related to the calculation and interpretation of FICI values. This limitation should be taken into account when extrapolating the findings. Moreover, future investigations should carefully consider factors that affect the reproducibility and reliability of peptide-based antimicrobial strategies. These include, but are not limited to, media composition, the purity of the synthesized peptides, the specific bacterial strains used in susceptibility testing, and other methodological details such as peptide formulation and assay conditions [55]. It should also be taken into account that certain solvents used for the preparation of peptide formulations, particularly DMSO, may exert membrane-thinning effects on cellular phospholipid bilayers, and at higher concentrations can induce pore formation, thereby increasing membrane permeability [56]. For example, in a study conducted on liposomes and erythrocyte membranes, DMSO at concentrations starting from 3% (*v*/*v*) was shown to alter the mechanical properties of the membrane and significantly increase its permeability [57]. In our study, the final DMSO concentration in all tested and control samples was relatively high at 2% (*v*/*v*), allowing for valid comparisons of antimicrobial effects between treated and control groups under identical solvent conditions. Nevertheless, the presence of DMSO in AMP formulations should be carefully considered in future development of therapeutic compositions in terms of both efficacy and safety. Ensuring the standardization and thorough reporting of these parameters will be essential for the validation and broader application of synergistic antimicrobial peptide combinations.

Overall, the results of this study support the use of antimicrobial peptides as promising candidates for combating bacterial infections, particularly in cases where synergy can be leveraged to enhance potency and reduce the effective concentrations required for bacterial inhibition. Future studies should further explore the mechanisms underlying these synergistic interactions and assess the in vivo efficacy of these peptide formulations.

## 4. Materials and Methods

### 4.1. Synthesis and Purification of Peptides

Peptides R23F^S^* (RKKRRQRRRGG-Sar-GVVVHI-X-GGKF-NH2), V31K^S^* (G-VVVHINGGKFGG-Sar-GSRQIKIWFQNRR-X-KWKK-NH2), and R44K^S^* (RKK-K-RQRRRGG-Sar-GVVVHINGGKFGG-Sar-GSRQIKIWFQNRR-X-KWKK-NH2) peptides were synthesized and verified for amino acid sequence accuracy as previously described in earlier studies [29,47]. Peptide bond formation was activated with TBTU, and coupling progress was monitored using Kaiser’s ninhydrin test. To address aggregation, acylation reactions were repeated using HFIP-DCM or THF-NMP solvent systems. Peptide deprotection and cleavage were performed with 1M TFMSA/thioanisole in TFA at 20 °C for 2 h. The crude peptides were precipitated with anhydrous ether, dried under vacuum, and neutralized with 0.1M aqueous NH_4_HCO_3_. Purification was carried out using Sephadex G-10 gel filtration followed by semi-preparative HPLC on a Luna C18 column (250 × 21.5 mm, 10 μm) at 10 mL/min, with 0.1% TFA in water (phase A) and acetonitrile (phase B). Fractions were analyzed by RP-HPLC on a Luna 5u C18 (2) 100 Å column (250 × 4.6 mm), lyophilized, and identified using an Orbitrap Elite mass spectrometer (Thermo Scientific, Dreieich, Germany). The observed molecular weights matched the calculated values, confirming successful synthesis (Appendix A).

### 4.2. Microorganism Strains

This study utilized the following strains: susceptible *Pseudomonas aeruginosa* (ATCC 28753), *Escherichia coli* (K12), *Bacillus cereus* (IP 5832), *Staphylococcus aureus* (209P), and the methicillin-, oxacillin-, and ampicillin-resistant MRSA strain (ATCC 43300).

### 4.3. Testing of Synergistic Antimicrobial Activity of Peptides Against E. coli, P. aeruginosa, MRSA, S. aureus, and B. cereus in a Liquid Medium and Calculation Fractional Inhibitory Concentration Indexes (FICIs)

The investigation of synergistic antimicrobial activity of peptides against bacterial strains including *Escherichia coli* (strain K12), *Pseudomonas aeruginosa* (strains ATCC 28753), methicillin-resistant *Staphylococcus aureus* (MRSA, strain ATCC 43300), *Staphylococcus aureus* (strain 209P), and *Bacillus cereus* (strain IP 5832) was conducted using a liquid medium, specifically Mueller-Hinton broth (MHB). Incubation conditions followed standard protocols previously established [58]. In our study, we combined the inhibitory concentration of one peptide with titrated concentrations of another peptide to calculate fractional inhibitory concentration indices. Gentamicin sulfate served as a positive control. Peptides and gentamicin were dissolved in 100% dimethyl sulfoxide (DMSO) to prepare test solutions containing final DMSO concentrations at 2% (*v*/*v*). The same concentration of DMSO was present in the bacterial cultures in the liquid medium that served as negative controls. Each bacterial strain along with tested peptides or antibiotic was incubated at 37 °C for 24 h, measuring the optical density every 30 min at a 600 nm wavelength on a Multiscan GO (Thermo Scientific, Waltham, MA, USA).

To evaluate potential additive or synergistic effects, the fractional inhibitory concentration index (FICI) was determined using Equation (1) [59]:(1)$$\mathrm{FICI}=\frac{MIC\;peptide\;1\;incombination}{MIC\;peptide\;1}+\frac{MIC\;peptide\;2\;in\;combination}{MIC\;peptide\;2}$$

Synergistic effects, characterized by a significantly enhanced antimicrobial activity compared to individual peptides, were defined by FICI values below 0.5. Additive effects, where the combination produced a greater effect than either peptide alone, were observed for FICI values ranging from 0.5 to 1. Values between 1 and 4 suggested no interaction between the AMPs. A FICI value greater than 4.0 indicated antagonism at the tested concentrations [59].

### 4.4. Statistical Analysis

Statistical analysis was performed using the SigmaPlot v14.5 software package (SPSS Inc., Chicago, IL, USA). Each experiment was conducted in two independent replicates, and data were expressed as mean ± standard deviation (M ± SD). The graphs were generated using OriginPro 2024 SR1 version 10.1.0.178 (OriginLab Corporation, Northampton, MA, USA).

## 5. Conclusions

Overall, our study highlights the potential of antimicrobial peptides as effective agents against bacterial infections, particularly when synergy enhances their potency and lowers the required concentrations for bacterial inhibition. Our findings demonstrate that the designed peptides exhibit significant antimicrobial activity, with their combinations producing synergistic and additive effects. These results support the growing interest in AMPs and CPP-modified peptides as alternatives to traditional antibiotics. Future research should focus on elucidating the mechanisms driving these interactions, optimizing peptide sequences for improved stability and activity, and evaluating their therapeutic potential in vivo. Additionally, expanding the range of tested bacterial strains and conducting detailed biophysical analyses will be crucial for refining these peptides for clinical applications.

## Figures and Tables

**Figure 1 ijms-26-05968-f001:**
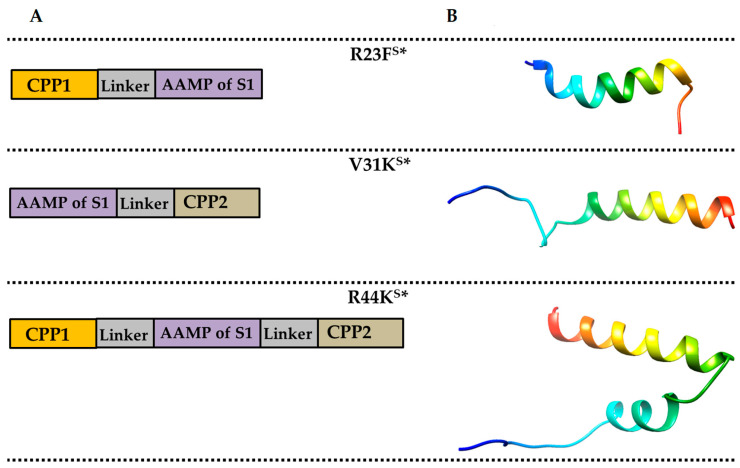
Peptide design scheme (**A**) and folding patterns of the peptides R23F^S^*, V31K^S^*, and R44K^S^* obtained using the AlphaFold 3 server (**B**) [30]. Designations: CPP1—fragment of the cell-penetrating peptide TAT, Linker—a sequence composed of glycine-glycine-sarcosine-glycine residues, AAMP of S1—a region containing an amyloidogenic fragment of the ribosomal protein S1 from *S. aureus* [29], CPP2—fragment of the cell-penetrating peptide Antp. The amino acid sequences of the peptides R23F^S^*, V31K^S^*, and R44K^S^* are provided in the Section 4.

**Figure 2 ijms-26-05968-f002:**
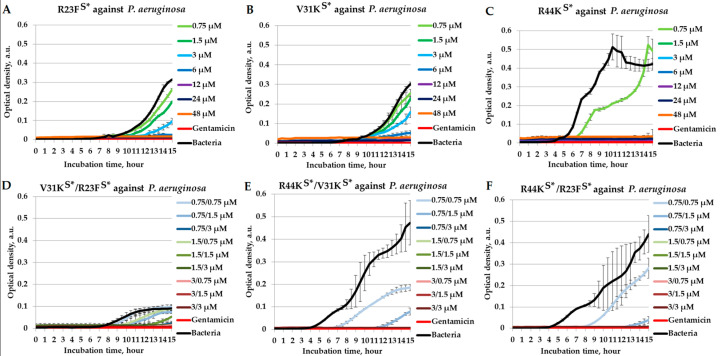
Combinatorial antimicrobial effects of peptides R23F^S^*, V31K^S^*, and R44K^S^* against *P. aeruginosa* (ATCC 28753 strain) in a liquid medium. The results are shown for different concentrations of the peptides R23F^S^* (**A**), V31K^S^* (**B**), and R44K^S^* (**C**) individually, as well as for the peptide combinations V31K^S^*/R23F^S^* (**D**), R44K^S^*/V31K^S^* (**E**), and R44K^S^*/R23F^S^* (**F**). Bacterial cultures in a liquid medium served as negative controls. Gentamicin sulfate was used as a positive control. Error bars show standard errors. The number of independent experiments is two.

**Figure 3 ijms-26-05968-f003:**
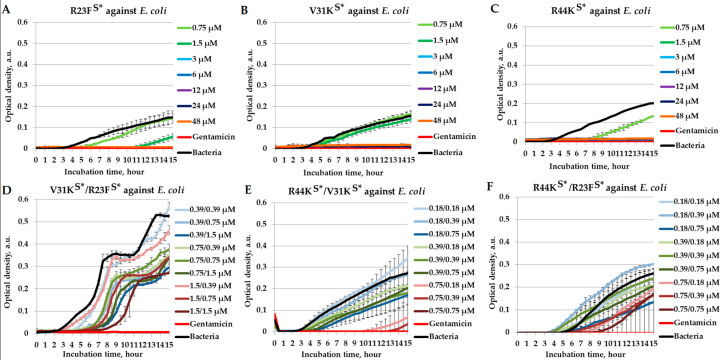
Combinatorial antimicrobial effects of peptides R23F^S^*, V31K^S^*, and R44K^S^* against *E. coli* (K12 strain) in a liquid medium. The results are shown for different concentrations of the peptides R23F^S^* (**A**), V31K^S^* (**B**), and R44K^S^* (**C**) individually, as well as for the peptide combinations V31K^S^*/R23F^S^* (**D**), R44K^S^*/V31K^S^* (**E**), and R44K^S^*/R23F^S^* (**F**). Bacterial cultures in a liquid medium served as negative controls. Gentamicin sulfate was used as a positive control. Error bars show standard errors. The number of independent experiments is two.

**Figure 4 ijms-26-05968-f004:**
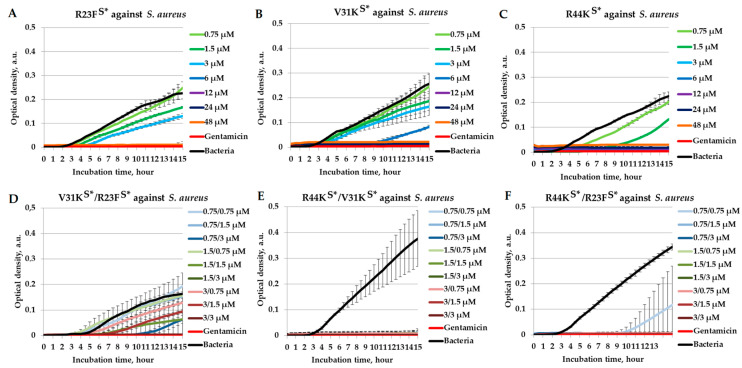
Combinatorial antimicrobial effects of peptides R23F^S^*, V31K^S^*, and R44K^S^* against *S. aureus* (209P strain) in a liquid medium. The results are shown for different concentrations of the peptides R23F^S^* (**A**), V31K^S^* (**B**), and R44K^S^* (**C**) individually, as well as for the peptide combinations V31K^S^*/R23F^S^* (**D**), R44K^S^*/V31K^S^* (**E**), and R44K^S^*/R23F^S^* (**F**). Bacterial cultures in a liquid medium served as negative controls. Gentamicin sulfate was used as a positive control. Error bars show standard errors. The number of independent experiments is two.

**Figure 5 ijms-26-05968-f005:**
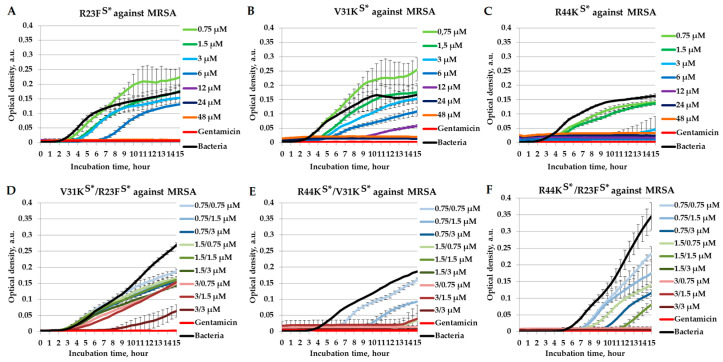
Combinatorial antimicrobial effects of peptides R23F^S^*, V31K^S^*, and R44K^S^* against MRSA (ATCC 43300 strain) in a liquid medium. The results are shown for different concentrations of the peptides R23F^S^* (**A**), V31K^S^* (**B**), and R44K^S^* (**C**) individually, as well as for the peptide combinations V31K^S^*/R23F^S^* (**D**), R44K^S^*/V31K^S^* (**E**), and R44K^S^*/R23F^S^* (**F**). Bacterial cultures in a liquid medium served as negative controls. Gentamicin sulfate was used as a positive control. Error bars show standard errors. The number of independent experiments is two.

**Figure 6 ijms-26-05968-f006:**
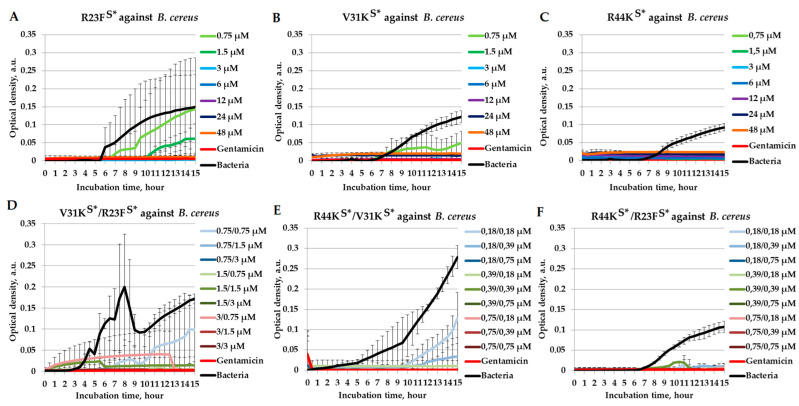
Combinatorial antimicrobial effects of peptides R23F^S^*, V31K^S^*, and R44K^S^* against *B. cereus* (IP 5832 strain) in a liquid medium. The results are shown for different concentrations of the peptides R23F^S^* (**A**), V31K^S^* (**B**), and R44K^S^* (**C**) individually, as well as for the peptide combinations V31K^S^*/R23F^S^* (**D**), R44K^S^*/V31K^S^* (**E**), and R44K^S^*/R23F^S^* (**F**). Bacterial cultures in a liquid medium served as negative controls. Gentamicin sulfate was used as a positive control. Error bars show standard errors. The number of independent experiments is two.

**Table 1 ijms-26-05968-t001:** The combinatorial antimicrobial activity of peptides, including fractional inhibitory concentration index (FICI) for antimicrobial combinations, against the tested bacterial species. MICs are expressed in µM.

Bacteria	Peptide 1	Peptide 2	Peptide 1 MIC in Combination	Peptide 1 MIC	Peptide 2 MIC in Combination	Peptide 2 MIC	FICI ^#^
*P. aeruginosa*	V31K^S^*	R23F^S^*	1.5	12	3	6	0.63
*E. coli*	V31K^S^*	R23F^S^*	No data	3	No data	3	No data
*S. aureus*	V31K^S^*	R23F^S^*	1.5	12	3	6	0.63
MRSA	V31K^S^*	R23F^S^*	No data	24	No data	12	No data
*B. cereus*	V31K^S^*	R23F^S^*	1.5	1.5	3	3	2
*P. aeruginosa*	R44K^S^*	V31K^S^*	0.75	1.5	3	12	0.75
*E. coli*	R44K^S^*	V31K^S^*	0.75	1.5	0.75	3	0.75
*S. aureus*	R44K^S^*	V31K^S^*	0.75	3	0.75	12	0.31
MRSA	R44K^S^*	V31K^S^*	3	6	0.75	24	0.53
*B. cereus*	R44K^S^*	V31K^S^*	0.18	0.75	0.75	1.5	0.74
*P. aeruginosa*	R44K^S^*	R23F^S^*	0.75	1.5	3	6	1
*E. coli*	R44K^S^*	R23F^S^*	No data	1.5	No data	3	No data
*S. aureus*	R44K^S^*	R23F^S^*	0.75	3	3	6	0.75
MRSA	R44K^S^*	R23F^S^*	1.5	6	3	12	0.5
*B. cereus*	R44K^S^*	R23F^S^*	0.18	0.75	0.75	3	0.49

^#^ FICI = $\frac{MIC\;peptide\;1\;in\;combination}{MIC\;peptide\;1}+\frac{MIC\;peptide\;2\;in\;combination}{MIC\;peptide\;2}$.

## Data Availability

Data is contained within the article and Appendix A.

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
