# Peer review of "Combinatorial Effects of CPP-Modified Antimicrobial Peptides: Synergistic and Additive Interactions Against Pathogenic Bacteria"

_ijms, 2025, doi:10.3390/ijms26135968_

Round 1
Reviewer 1 Report
Comments and Suggestions for Authors
The manuscript offers a well-organized examination of the combined effects of CPP-modified antimicrobial peptides, concentrating on synergistic and additive interactions against various clinically significant pathogens. The research is well-designed and supported by suitable data. To enhance transparency of the work, clarity, and methodological rigor, the following suggestions should be considered:
Although the authors mention purification through semi-preparative HPLC and analysis via RP-HPLC, they do not provide any purity values or chromatograms to support these methods.
Given the concentration-dependent antimicrobial effects, it is crucial to report the final peptide purity (%) to establish that the biological outcomes can be attributed only to the intended peptide sequence. Could you please address the inconsistencies in the reporting of replicates? There are contradictory statements throughout the manuscript regarding the number of replicates conducted. The authors state in Section 4.3 that all experiments were performed in triplicate, implying a sample size of n = 3. However, in Section 4.6, they state that each experiment was conducted at least twice (n ≥ 2), suggesting possible variability. Notably, the figure legends for Figures 1–5 uniformly state that the number of independent experiments is two, which denotes n = 2 biological replicates. These inconsistencies may raise concerns about the reproducibility and statistics of the conclusions. Please clarify whether the mentioned triplicates refer to technical or biological replicates, and revise the manuscript for uniformity. If n = 2 was used across all conditions, this should be indicated and acknowledged as a limitation in the discussion, particularly when interpreting FICI-based outcomes.
The manuscript utilizes 2% DMSO as a solvent for peptide and antibiotic delivery, consistent with the authors’ earlier publication (IJMS 2021, doi:10.3390/ijms22189776). I reviewed both studies to confirm, and neither the study nor the current manuscript includes a discussion of DMSO’s known effects on bacterial membrane permeability or antimicrobial assay sensitivity, particularly regarding strains such as E. coli, P. aeruginosa, and S. aureus. To enhance transparency, I suggest that the authors briefly address this limitation in the discussion section. Specifically, please clarify whether DMSO-alone controls were conducted and consider referencing studies that discuss the membrane-modulatory effects of DMSO at similar concentrations (≥1%). This would help ensure that solvent effects don't influence the observed peptide activity and results.
Finally, the manuscript mentions multifunctional peptides that include cell-penetrating sequences (TAT, Antennapedia) and amyloidogenic fragments; however, these modular components are presented as complete sequences. A schematic illustration of peptide design, showing the arrangement of functional domains and key modifications (for example, R44K substitution), would enhance clarity of this aspect, especially for readers unfamiliar with the design strategy. This would also help in understanding the structure–activity relationships detailed in the results.
Reviewer 2 Report
Comments and Suggestions for Authors
In this work, the authors investigate the synergistic effects observed when using binary hybrid peptide treatments. The work is interesting and provides important information for the design of new biological control strategies. However, the text should include in the introduction a little more information about the mechanisms of action of CPPs and AMPs, to better understand the synergistic effects reported in this study. Additionally, the manuscript fails to provide the structure of the peptides used. Although the sequence information for each peptide is presented in the Methods section, the interpretation of the results is difficult without a clear structure for each peptide. Rather, the authors prefer to refer to previous works where this information is shown, which does not always work for a wide group of readers interested in this topic.
One of the main contributions of this work is the finding of the R44K S*/V31K S* treatment, which was highly effective against S. aureus but not so much against MRSA. However, the authors do not sufficiently emphasize this point and leave aside a broader discussion regarding the differences of these two types of strains, in terms of metabolism, physiology, or lipid composition, which could also offer important information for combating this class of pathogens.
Finally, authors should be reminded that specific names (e.g., E. coli, S. aureus, B. cereus) should not be capitalized, as this is a common omission in manuscripts. In addition, an n = 2 for the experiments is not usually the most common, so it is recommended to include a third repetition for each experiment or to indicate in which cases this has been done but it has been decided not to show it.
